# Judgement Differences of Types of Image-Based Sexual Harassment and Abuse Conducted by Celebrity Perpetrators and Victims

**DOI:** 10.3390/bs14111021

**Published:** 2024-11-01

**Authors:** Dean Fido, Alex Rushton, Ellie Allen, Jackie Williams

**Affiliations:** College of Health, Psychology and Social Care School of Psychology Kedleston Road, University of Derby, Derby DE22 1GB, UK

**Keywords:** image-based sexual harassment and abuse, unsolicited sending of intimate images, non-consensual sharing of intimate images, victim blame, psychopathy, proclivity

## Abstract

An emerging corpus exists pertaining to societal judgements of image-based sexual harassment and abuse (IBSHA). This type of research centres on the non-consensual sharing of intimate images (NCSII; sometimes called ‘revenge pornography’), but recent legislative developments seeking to convict those who engage in the unsolicited sending of intimate images (USII; sometimes called ‘dick pics’) evoke a need to broaden this literature. Moreover, in the context of recent and highly publicised accounts featuring both celebrity perpetrators and victims of IBSHA, it is important to understand whether celebrity status impacts said judgements. We present three studies outlining judgement differences between vignettes featuring NCSII and USII as a function of perpetrator/victim celebrity status and as predicted by previously implicated personality traits and beliefs. In Studies 1 (N = 261) and 2 (N = 237), though vignettes involving NCSII were perceived more criminal in nature and anticipated to evoke more harm than USII, said victims also received more blame. Contrary to our hypotheses, there was no further impact of celebrity status on either the perpetrator (Study 2) or victim (Study 3; N = 207). Finally, although dark personality traits (associated with callousness and low empathy) predicted variation in judgements of IBSHA across all studies, in Study 2, psychopathic personality traits specifically predicted proclivity to engage in NCSII but not USII. The results are discussed in reference to the importance of IBSHA-related education on an international level and the pursuit of further legislation in this area.

## 1. Introduction

Image-based sexual harassment and abuse (IBSHA) represents a constellation of behaviours encompassing the unsolicited sending (USII) and non-consensual capture (e.g., upskirting), creation (e.g., deepfake sexual media production), and sharing (NCSII, i.e., so-called ‘revenge pornography’) of intimate images or videos [1,2,3]. Despite NCSII being criminalised in the United Kingdom (UK) since 2015 (Criminal Justice and Courts Act, 2015 [4]) and recent legislative developments being implemented to safeguard victims of upskirting, deepfaking, and USII (Online Safety Act, 2023 [5]; Voyeurism (Offences) Act, 2019 [6]), IBSHA remains a global problem [7] that is in need of immediate psycho-legal investigation.

In the context of individuals often being unaware of their victimisation [8], existing fears of re-victimisation from reporting [1], and inconsistent recording methods [9], the Revenge Porn Helpline [10] reported a 106% increase in service referrals between 2022 and 2023, with empirical data suggesting around one-third of people have experienced some form of IBSHA [11]. Comparable data for USII is less concrete, but one dataset from Canada suggests that 68% of their college student sample reported having received at least one unsolicited sexual image throughout their lives [12]. Though some evidence suggests men experience comparable levels of NCSII to women [13], IBSHA more broadly is typically considered a gendered crime [14,15], wherein victims experience social (e.g., relationship breakdowns, reputational damage; [16]) and professional (e.g., employment termination; [17]) consequences in addition to increased depression, anxiety, and suicidal ideation [3,18]. Victims of USII report experiencing anger and disrespect [19,20].

Despite this, victims of IBSHA are frequently attributed blame during their victimisation, such as for creating and sharing sexual images in the first instance [7], akin to what is seen in physical sexual abuse cases, with victims blamed for being drunk or dressing provocatively. For example, Henry et al. [21] indicate pervasive societal attitudes that minimise NCSII-related impacts, with 55.2% [22] and 71% [23] of samples attributing at least some responsibility to the victim. Such beliefs are likely fuelled by a marked increase in the use of dating apps to develop and maintain sexual relationships [24] and the sending of sexual images being considered a societal norm [25,26]. However, we have yet to know whether such findings are mirrored in the context of USII. Though one might expect more victim-supporting judgements owing to them unlikely having previously engaged with the perpetrator, a competing hypothesis might be that owing to legislation only recently (in the UK) being passed to govern USII and therefore low societal education on its associated harms, more lenient judgements might be observed compared with those given to cases involving NCSII. Regardless, research from the social sciences suggests that IBSHA-related judgements might be further predicted by variations in personality traits and prior beliefs.

### 1.1. Predictors of IBSHA-Related Judgements

So-called *dark personality* traits (i.e., psychopathy, narcissism, Machiavellianism) are characterised by low empathic responses, inappropriate affective responses, and a need for immediate sexual gratification [27]. Although all have clinical relevance, they exist on a continuum within the general population [28] and have been associated with deviant online (e.g., harassment, trolling, exploitation) and offline (e.g., pro-rape beliefs, gender-based violence, infidelity-induced revenge seeking) behaviour [23,29,30,31,32].

Though unsurprising that such traits are frequently investigated alongside IBSHA judgements, conflicting findings are presented. For example, Fido’s group found psychopathy and subscales thereof to predict more lenient judgements of both NCSII [33,34,35] and deepfake media production [36], but they were unable to replicate this in upskirting cases [34]. Moreover, despite Pina et al. [37] positively associating enjoyment of NCSII with narcissism and Machiavellianism, their later work suggested that this did not extend to NCSII proclivity [23]. However, this was not the case in either Swanek [38] or Karasavva and Forth [11] where psychopathy and narcissism were positively associated with self-reported NCSII, nor in Morelli et al. [39], where psychopathy and Machiavellianism predicted engagement in USII. Thus, work is required to explore methodological and cultural variations presented within this corpus.

Worldly beliefs, such as ‘just world’ thinking [40], where one assumes that the world is fair and that people get what they deserve [41], also predict IBSHA-related judgements, wherein this assignment of responsibility helps to rationalise and remove any distress the observer may feel about experiencing victimisation themselves [42]. Over a series of studies in the context of NCSII, Harper et al. [35] found beliefs about a just world (BJWs) to be associated with offense minimisation, viewing victims as being promiscuous and engaging in avoidable behaviour, views that NCSII resulted in minimal harm and was not criminal in nature, and proclivity to engage in NCSII. Though not yet explored in the context of USII, we might expect similar findings given that these results were comparable in the contexts of deepfake media production [36] and both domestic and sexual abuse cases [43,44] where no victim indiscretions were showcased. Though it has yet to be explored empirically, beliefs about gender roles might also predict IBSHA-related judgement scores, mapping on to viewpoints reinforcing men’s drive for patriarchal dominance via victimising women, as well as viewing them as objects for sexual gratification more broadly [3,9]. Such views are prevalent in domestic and sexual abuse scenarios (e.g., [45,46,47] and may—to some extent—be supported by Harper et al. [35], who found right-wing values as being associated with anti-victim judgements in NCSII contexts.

Finally, though international comparisons of NCSII judgements have been performed (e.g., [48]), specific exploration of beliefs about one’s culture in the context of IBSHA is lacking. Historically, and on a societal level, individuals from collectivist cultures (e.g., Singapore), wherein community needs are valued over and above those of the individual, are more punitive [49,50] and assign greater blame to both perpetrators [51] and victims of crime [52] than individuals from individualistic cultures (e.g., UK, America). Moreover, such blame also extends to one’s broader social circles [51]. Of course, it is possible that one might prescribe to a given perspective but live in a society that broadly contradicts such values [53]. In the context of IBSHA, it is therefore likely that individuals from collectivist cultures will prescribe greater blame across all those involved in the behaviour. Moreover, individuals from collectivist cultures may deem said actions to be more criminal in nature than those from individualistic cultures, especially if they also prescribe to collectivist values themselves.

### 1.2. The Role of Celebrity Status in IBSHA Judgements

Despite the above, there are also social factors that should be considered. When celebrities are positioned as either perpetrators or victims of domestic (e.g., Johnny Depp, Amber Heard) or sexual (e.g., Harvey Weinstein, Angelina Jolie) abuse, said cases rapidly foster public interest [54]. Though such publicity can help victims understand their own experiences [55], they can also facilitate surges in rape myth acceptance, misogyny, and victim blame [56]. More recently, cases of IBSHA featuring celebrities have also been featured in the public domain, mapping on to the long-conceded understanding that female celebrities are disproportionately targeted by IBSHA [57]. Stephen Bear was sentenced to 21 months of imprisonment in 2023 for profiting off of NCSII featuring then-partner and fellow television personality, Georgia Harrison, and former actor Laurence Fox underwent police investigation in 2024 for posting an archived upskirting image of television presenter Narinder Kaur. In the context of USII, despite there being no convictions to date, actress Emily Atack and television personality Pete Wicks have both publicly commented on their receipt of USII [58]. Such actions might represent entitled beliefs [59] and atypical relationship formation strategies [2].

To date, only one study has empirically explored the impact of celebrity status on judgements of IBSHA [36], wherein celebrity victims were blamed more and perceived to be impacted less by their victimisation relative to non-celebrities. This was especially the case when said victims were male. Noteworthy, this study was limited to explorations of celebrity victims (not perpetrators) in the context of deepfaked sexual media production (not NCSII nor USII).

### 1.3. Overview of Studies

Owing to the need to better understand unique differences in judgements of and proclivities to commit NCSII and USII, we present three individually sampled studies, which utilise cross-sectional designs with written vignettes as experimental manipulations. In Study 1, we tested baseline judgement differences between NCSII and USII as a function of the victim’s sex. Then, in Studies 2 and 3, we extended this work to explore whether judgements differed as a function of the perpetrator and/or victim having celebrity status, respectively. Moreover, in Study 3, we also examined whether judgements of NCSII in particular differed from those awarded to sexual abuse or domestic abuse (not featuring sexual elements). We also sought to predict said judgements via psychological traits, worldly beliefs, demographic factors, and prior IBSHA-related victimisation. We predicted that vignettes involving USII would be viewed less punitively than those featuring NCSII, which in turn would be viewed less punitively than vignettes featuring either sexual abuse or domestic abuse. Moreover, a typology of participants comprising older males, with no prior history of IBSHA-related victimisation and who self-report greater psychopathic personality traits, beliefs about a just world and gender norms, and collectivist values, was predicted to report harsher victim judgements.

## 2. Study 1

### 2.1. Methods

#### Participants

For each study, a priori analyses (G*Power version 3.1.9.2) determined target sample sizes. Anticipating small-to-medium effects (standard alpha = 0.05) between 220 and 274 participants were required to have 80% power. All participants were 18 years of age or over, fluent in English, and UK-based to control for variance in legislation, and they were recruited through social media (e.g., Reddit, *X*, Facebook). For Study 1, after removing instances where >5% of data were missing (*n* = 5), 261 datasets were analysed (*M_age_* = 32.99 years, *SD* = 14.19; 49.4% female). Of these participants, 6.1% (*n* = 16) and 13.4% (*n* = 35) reported having been a victim or knowing a victim of NCSII, respectively, and 26.1% (*n* = 68) and 31.8% (*n* = 85) reported having been a victim or knowing a victim of USII, respectively.

### 2.2. Materials

**Judgements of Image-Based Sexual Abuse:** Judgements were measured using a vignette and scale design, which was first featured in Bothamley and Tully [60] before being adapted in Fido et al. [33,34,36]. Participants read one of four vignettes, which differed as a function of [i] whether they featured NCSII or USII; and [ii] the sex of the victim (only heterosexual pairings were used). Vignettes for all studies are available here: https://osf.io/27uty/?view_only=03c41479e43e495fa5ad1b35762ddb9d, accessed on 27 September 2024.

After affirming that they had read and understood the vignette, participants answered eight judgement-related items comprising three subscales: *victim blame* (e.g., ‘*How much do you think that (victim’s name) is to blame for the incident?*’; α = 0.762; 4 items), *victim harm* (e.g., ‘*Do you thinks (perpetrator’s name) behaviour will cause harm to (victim’s name)?*’; α = 0.741; 2 items), and *perceived criminality* (e.g., ‘*Do you think police intervention is necessary for resolution of this situation?*’; α = 0.904; 2 items). Responses were scored on a 7-point Likert scale, anchored from 1 (‘*not at all/very unlikely*’) to 7 (‘*definitely/very likely*’), with higher averaged scores reflecting stronger beliefs.

**The Short Dark Triad (SD3; [28]):** The SD3 is a 27-item measure of dark personality comprising three subscales: *psychopathy* (e.g., ‘*Payback needs to be quick and nasty*.’; α = 0.747; 9 items), *narcissism* (e.g., ‘*I know that I am special because everyone keeps telling me so.*’; α = 0.725; 9 items), and *Machiavellianism* (e.g., ‘*I like to use clever manipulation to get my own way.*’; α = 0.718; 9 items). Responses were recorded on a 5-point Likert scale ranging from 1 (‘*Strongly disagree*’) to 5 (‘*Strongly agree*’), with higher averaged scores reflecting greater resonance with the given trait.

**Individualism and Collectivism Scale (ICS; [53]):** The ICS is a 16-item measure of one’s cultural orientation composed of four subscales: *horizontal individualism* (HI, e.g., ‘*I’d rather depend on myself than others*’; α = 0.749; 4 items), *vertical individualism* (VI, e.g., ‘*Winning is everything*’; α = 0.778; 4 items), *horizontal collectivism* (HC, e.g., ‘*I feel good when I cooperate with others*’; α = 0.799; 4 items), and *vertical collectivism* (VC, e.g., ‘*Parents and children must stay together as much as possible*’; α = 0.81; 4 items). Responses were recorded on a 9-point Likert scale ranging from 1 (‘*Never/definitely no*’) to 9 (‘*always/definitely yes*’), with greater scores reflecting greater value alignment.

**Incidence of Victimisation:** Participants opted (all but one obliged) to report if someone within their social circle and/or themselves had ever had their intimate images disseminated without consent (i.e., NCSII) or received unsolicited sexual images (i.e., USII).

### 2.3. Procedure

After clicking the study link and providing consent via Qualtrics, participants answered demographic questions. Next, they were randomly allocated to one of four groups, which determined the vignette they viewed. After reading the vignette and answering the judgement questions, they completed the SD3 and ICS, which were randomly presented to mitigate order effects. Finally, participants answered victim-related measures before being debriefed. This procedure was approved by an institutional ethics committee [REF: ETH2324-0702].

### 2.4. Results

#### How Does Victim Sex and Modality of Offence Affect Judgements?

Data for all studies were analysed in SPSS (version 28). After screening for outliers and undertaking assumption-related processes, three 2 (offence modality: NCSII vs. USII) × 2 (victim sex: male vs. female) ANCOVA tests determined the main and interaction effects on the judgement scores (victim blame, perceived criminality, anticipated harm) whilst controlling for prior self-victimisation. Descriptive statistics are presented in Table 1.

***Victim Blame:*** There was a significant main effect of modality on victim blame, *F*(1, 254) = 168.79, *p* < 0.001, η^2^ = 0.399, whereby greater blame was attributed in cases of NCSII (*M* = 3.57, *SD* = 1.29) than USII (*M* = 1.78, *SD* = 0.90). There was neither a main effect of victim sex, *F*(1, 254) = 0.64, *p* = 0.425, η^2^ = 0.003, nor interaction thereof, *F*(1, 254) = 0.19, *p* = 0.665, η^2^ = 0.001. Only the covariate of one’s own victimisation of USII, *F*(1, 254) = 7.70, *p* = 0.006, η^2^ = 0.029, but not NCSII, *F*(1, 254) = 0.32, *p* = 0.574, η^2^ = 0.002, statistically (and positively) predicted victim blame.

***Perceived Criminality:*** There was a significant main effect of modality on perceived criminality, *F*(1, 254) = 46.01, *p* < 0.001, η^2^ = 0.153, whereby greater perceived criminality was observed for NCSII (*M* = 4.34, *SD* = 1.49) than USII (*M* = 3.13, *SD* = 1.44). There was also a significant main effect of victim sex, *F*(1, 254) = 33.25, *p* <.001, η^2^ = 0.116, whereby greater perceived criminality was present in vignettes depicting female (*M* = 4.27, *SD* = 1.50) relative to male victims (*M* = 3.24, *SD* = 1.49). The interaction was not statistically significant *F*(1, 254) = 1.92, *p* = 0.167, η^2^ = 0.007, nor were the covariates of self NCSII, *F*(1, 254) = 0.04, *p* = 0.851, η^2^ = 0.000, or USII victimisation, *F*(1, 254) = 0.48, *p* = 0.490, η^2^ = 0.002.

***Anticipated Harm:*** There was a significant main effect of modality on anticipated harm, *F*(1, 254) = 146.45, *p* < 0.001, η^2^ = 0.366, whereby higher anticipated harm was observed for NCSII (*M* = 6.04, *SD* = 1.04) than USII (*M* = 4.16, *SD* = 1.52). There was also a significant main effect of victim sex, *F*(1, 254) = 32.13, *p* < 0.001, η^2^ = 0.112, whereby higher anticipated harm was present in vignettes depicting female (*M* = 5.60, *SD* = 1.29) relative to male victims (*M* = 4.66, *SD* = 1.74). The interaction was not statistically significant, *F*(1, 254) = 3.32, *p* = 0.070, η^2^ = 0.013, nor were the covariates of self NCSII, *F*(1, 254) = 0.89, *p* = 0.348, η^2^ = 0.003, or USII victimisation, *F*(1, 254) = 0.01, *p* = 0.977, η^2^ = 0.000.

### 2.5. To What Extent Does Psychological Traits Predict Judgements of IBSHA?

Three multiple regression models were computed for each vignette, one for each judgement outcome, through which the psychometric variables were entered as predictors. The regression values for each model are presented in Table 2, alongside standardised estimates to aid clarity.

**NCSII:** When the victim was female, the model for victim blame, R^2^adj = 0.047, *F*(9, 59) = 1.37, *p* = 0.220, was not statistically significant, but models for perceived criminality, R^2^adj = 0.163, *F*(9, 59) = 2.47, *p* = 0.018, and anticipated harm, R^2^adj = 0.155, *F*(9, 59) = 2.39, *p* = 0.022, were. Lower vertical individualism (β = 0.311, *p* = 0.027) and higher horizontal collectivism (β = 0.316, *p* = 0.044) predicted greater perceived criminality, and both higher Machiavellianism (β = 0.305, *p* = 0.030) and horizontal collectivism (β = 0.375, *p* = 0.018) predicted greater anticipated harm. When the victim was male, the models for victim blame, R^2^adj = 0.284, *F*(9, 55) = 3.82, *p* < 0.001, and anticipated harm, R^2^adj = 0.299, *F*(9, 55) = 4.04, *p* < 0.001, were statistically significant. Being older (β = 0.568, *p* < 0.001) and having higher Machiavellianism (β = 0.361, *p* = 0.023) predicted greater victim blame, and being younger (β = −0.423, *p* = 0.002) predicted greater anticipated harm. Though the model for perceived criminality was statistically significant, R^2^adj = 0.147, *F*(9, 55) = 2.23, *p* = 0.034, no individual predictor was.

**USII:** When the victim was female, neither model for victim blame, R^2^adj = 0.137, *F*(9, 50) = 2.04, *p* = 0.054, perceived criminality, R^2^adj = 0.007, *F*(9, 50) = 1.05, *p* = 0.416, or anticipated harm, R^2^adj = 0.017, *F*(9, 50) = 1.11, *p* = 0.371, were statistically significant. The same was found for when the victim was male for victim blame, R^2^adj = 0.046, *F*(9, 57) = 1.36, *p* = 0.230: perceived criminality (R^2^adj = 0.056, *F*(9, 57) = 1.44, *p* = 0.194, and anticipated harm (R^2^adj = 0.076, *F*(9, 57) = 1.61, *p* = 0.136).

## 3. Study 2

Study 2 sought to both replicate the variation in judgements seen in Study 1 as a function of offence modality (i.e., NCSII vs. USII) as well as explore whether said judgements varied further as a function of the perpetrator status (i.e., celebrity vs. non-celebrity). Moreover, to further build on this corpus of knowledge, we delineated participants’ proclivity to commit NCSII and USII as a function of *dark* personality traits.

### 3.1. Methods

#### Participants

After removing instances of >5% missing data (*n* = 6), 237 participants (*M*_age_ = 24.76 years, *SD* = 6.37; 76.8% female) were retained.

### 3.2. Materials

**The Short Dark Triad:** The SD3 scale was used in line with Study 1: narcissism (α = 0.680), psychopathy (α = 0.741), Machiavellianism (α = 0.753).

**Judgements of Image-Based Sexual Abuse:** Participants read one of four randomly presented vignettes differing as a function of [i] whether they featured instances of NCSII or USII; and [ii] the status of the perpetrator (celebrity vs. non-celebrity). In line with Fido et al. [36], the non-celebrity perpetrator was a barista. Judgement items followed: *victim blame* (α = 0.803), *perceived criminality* (α = 0.557), and *anticipated harm* (α = 0.803).

**Proclivity to Engage in IBSHA:** Participants’ proclivity to engage in NCSII and USII were assessed using two single-item percentage-based scales, ranging from 0% (‘*Extremely unlikely*’) to 100% (‘*Extremely likely*’). In line with Fido et al. [34,36], these items were prefixed with the context of ‘*If there was 0% chance of being caught, what is the likelihood that you would…*’ (1) ‘share a sexually explicit image of somebody with someone else without their consent?’ and (2) ‘send an unsolicited sexual image to someone?’.

#### Procedure

An identical procedure was adopted to Study 1, save for proclivity measures included prior to debriefing. This procedure was ethically approved [REF: ETH2223-2028].

### 3.3. Results

#### How Does Perpetrator Status and IBSHA-Type Effect Judgements?

We ran a series of 2 (perpetrator status: non-celebrity vs. celebrity) × 2 (offence modality: NCSII vs. USII) between-groups ANOVAs, with the descriptive statistics being presented in Table 3. The dependent variables were victim blame, perceived criminality, and anticipated harm.

**Victim Blame:** There was a significant main effect of modality on victim blame, *F*(1, 233) = 192.681, *p* < 0.001, η^2^ = 0.453, whereby greater blame was attributed in cases of NCSII (*M* = 3.451, *SE* = 0.103) than USII (*M* = 1.433, *SD* = 0.103). There was neither a significant main effect of perpetrator status, *F*(1, 233) = 0.364, *p* = 0.547, η^2^ = 0.002, nor interaction thereof, *F*(1, 233) = 2.430, *p* = 0.120, η^2^ = 0.010.

**Perceived Criminality:** There was neither a significant main effect of modality, *F*(1, 233) = 3.297, *p* = 0.071, η^2^ = 0.014, perpetrator status, *F*(1, 233) = 0.974, *p* = 0.325, η^2^ = 0.004, nor interaction thereof, *F*(1, 233) = 0.836, *p* = 0.362, η^2^ = 0.004, on perceived criminality.

**Anticipated Harm:** There was a significant main effect of modality on anticipated victim harm, *F*(1, 233) = 35.458, *p* < 0.001, η^2^ = 0.132, whereby greater anticipated harm was attributed in cases of NCSII (*M* = 6.153, *SE* = 0.108) than USII (*M* = 5.249, *SE* = 0.107). There was no main effect of perpetrator status, *F*(1, 233) = 0.980, *p* = 0.323, η^2^ = 0.004 nor interaction thereof, *F*(1, 233) = 0.487, *p* = 0.486, η^2^ = 0.002.

### 3.4. How Do Demographics and Personality Traits Predict Judgements of IBSHA?

We ran a linear regression model for each outcome variable across each vignette, where psychometric and demographic variables were entered as predictors. Regression values are presented in Table 4 and feature standardised estimates.

**NCSII:** When the perpetrator was a non-celebrity, neither of the models for victim blaming, R^2^adj = 0.099, *F*(5, 55) = 2.313, *p* = 0.056, perceived criminality, R^2^adj = −0.033, *F*(5, 55) = 0.619, *p* = 0.686, or anticipated harm, R^2^adj = 0.026, *F*(5, 55) = 1.314, *p* = 0.272, were statistically significant. When the perpetrator was a celebrity, the model explaining victim blaming, R^2^adj = 0.159, *F*(5, 51) = 3.116, *p* = 0.016, was statistically significant, driven by higher Machiavellianism (β = 0.503, *p* = 0.006) and *lower* psychopathy (β = −0.368, *p* = 0.039). Neither the models of perceived criminality, R^2^adj = −0.004, *F*(5, 51) = 0.958, *p* = 0.452, nor anticipated harm, R^2^adj = 0.007, *F*(5, 51) = 1.084, *p* = 0.380, were statistically significant.

**USII:** When the perpetrator was a non-celebrity, models for victim blaming, R^2^adj = 0.137, *F*(5, 52) = 2.812, *p* = 0.025, perceived criminality, R^2^adj = 0.110, *F*(5, 52) = 2.404, *p* = 0.049, and anticipated harm, R^2^adj = 0.121, *F*(5, 52) = 2.572, *p* = 0.037, were statistically significant. Higher psychopathic traits (β = 0.318, *p* = 0.027) and being older (β = 0.367, *p* = 0.008) predicted greater victim blaming, and lower narcissism predicted greater perceived criminality (β = −0.402, *p* = 0.009) and anticipated harm (β = −0.385, *p* = 0.012). When the perpetrator was a celebrity, neither of the models for victim blaming, R^2^adj = 0.100, *F*(5, 55) = 2.339, *p* = 0.054, perceived criminality, R^2^adj = 0.033, *F*(5, 55) = 1.408, *p* = 0.236, nor anticipated harm, R^2^adj = 0.021, *F*(5, 55) = 1.259, *p* = 0.295, were statistically significant.

Can Demographics and Psychological Traits Predict Proclivity to Engage in IBSHA?

Two linear regressions outlined relationships between personality and demographic factors and proclivities for engaging in NCSII and USII. Though the composite NCSII model was not statistically significant, R^2^adj = 0.006, *F*(5, 226) = 1.278, *p* = 0.274, an increased proclivity to commit NCSII was associated with higher psychopathic traits (β = 0.189, *p* = 0.017). The USII model was statistically significant, R^2^adj = 0.060, *F*(5, 225) = 3.946, *p* = 0.002, wherein USII was only associated with being older (β = 0.135, *p* = 0.041) and male (β = −0.171, *p* = 0.011). Standardised coefficients are provided in Table 5.

## 4. Study 3

Study 3 builds on Study 2 three-fold: first, by transitioning from understanding variation in judgements as a function of perpetrator status to investigating *victim status* (i.e., celebrity vs. non-celebrity); second, by transitioning from differentiating between modality in the form of NCSII and USII to NCSII and situations depicting more physical manifestations of abuse, namely domestic abuse (featuring physical, but not sexual abuse) and sexual abuse specifically; third, throughout Studies 1 and 2, a distinct lack of association between psychopathy and judgements of IBSHA that have comprehensive been discussed elsewhere (e.g., [28,33,41]) was noted. As such, this study explores whether this lack of association represents a shift in the literature or reflects the SD3 as being limited in measuring psychopathy (nine items, composite measure) versus a more comprehensive measure.

### 4.1. Methods

#### Participants

After removing instances of >5% missing data (*n* = 16), 207 participants (*M*_age_ = 26.88, *SD* = 9.40, 79% female) were retained.

### 4.2. Materials

**Judgements of Image-Based Sexual Abuse:** Using the same approach described in Studies 1 and 2, participants read one of six randomly presented vignettes which differed as a function of: [i] whether they featured NCSII, domestic abuse (DA) with no sexual element, or sexual abuse (SA), and [ii] the status of the victim (celebrity vs. non-celebrity). The non-celebrity victim was depicted as a business consultant. The judgement items followed: *victim blame* (α = 0.724), *perceived criminality* (α = 0.772), and *victim harm* (α = 0.876).

**Self-Report Psychopathy Scale v.4 (short form) (SRP4; [61]):** The SRP4 measured psychopathic personality through 29 items across four subscales: *affective* (e.g., ‘Most people are wimps’; α = 0.730; 7 items), *interpersonal* (e.g., ‘I would get a kick out of ‘scamming’ someone’; α = 0.782; 7 items), *antisociality* (e.g., ‘I have never been involved in delinquent gang activity’; α = 0.611; 8 items) and *lifestyle* (e.g., ‘I’m a rebellious person’; α = 0.726; 7 items). Responses were recorded on a 5-point Likert scale ranging from 1 (‘*Strongly disagree*’) to 5 (‘*Strongly agree*’), with higher total scores indicating higher psychopathic personality traits.

**General and Personal Belief in Just World Scale (BJW; [62]):** The BJW measured one’s agreement that people get what they deserve in life through 13 items across *general* (e.g., ‘I think basically the world is a just place’; α = 0.750; 6 items) and *personal* (e.g., ‘overall, events in my life are just’; α = 0.869; 7 items) beliefs. Responses were recorded on a 6-point Likert scale ranging from 1 (‘*Strongly disagree*’) to 6 (‘*Strongly agree*’), with higher total scores reflecting stronger beliefs.

**Gender Role Belief Scale (short version) (GRBS; [63]):** The GRBS measured beliefs about gender norms through 10 items across two sets of beliefs: those about *women’s roles in households and workplaces* (e.g., ‘Women should have as much sexual freedom as men’; α = 0.748; 5 items) and those about *protectionism and chivalry toward women* (e.g., ‘It is disrespectful to swear in the presence of a lady’; α = 0.723; 5 items). Responses were recorded on a 7-point Likert scale ranging from 1 (‘*Strongly agree*’) to 7 (‘*Strongly disagree*’), with higher total scores indicating low gender role beliefs, which endorse feminism and dispute masculine and feminine gender constructs.

**Victim Status**: Participants opted (all obliged) to report if they themselves had ever had their intimate images disseminated without their consent (i.e., NCSII) or considered themselves to have faced domestic and/or sexual abuse.

### 4.3. Procedure

The procedure from Studies 1 and 2 was adopted, save for judgement measures being followed by the BJW, SRP4, and GRBS, in a randomised order to mitigate order effects. The victim status question was asked prior to debriefing. This procedure was ethically approved [REF: ETH2324-1722].

### 4.4. Results

#### 4.4.1. How Does Victim Status Affect Judgements of Offence Types?

We ran a series of 2 (victim status: celebrity vs. non-celebrity) × 3 (offence modality: NCSII vs. DA vs. SA) between-groups ANCOVAs, controlling for participants’ age, sex, and prior victimisation. The dependant variables were *victim blame*, *perceived criminality*, and *anticipated harm*. Descriptive statistics for each vignette are presented in Table 6.

**Victim Blame:** There was a significant main effect of modality, *F*(2, 198) = 4.968, *p* = 0.008, η^2^ = 0.048, whereby greater blame was attributed in cases of NCSII (*M* = 2.194, *SE* = 0.110) than SA (*M* = 1.704, *SE* = 0.112). No significant differences were present between DA (*M* = 1.907, *SE* = 0.106) and either NCSII or SA. There was neither a main effect of victim status, *F*(1, 198) = 0.657, *p* = 0.418, η^2^ = 0.003, nor the interaction thereof, *F*(2, 198) = 2.483, *p* = 0.086, η^2^ = 0.024. Past victimisation *F*(1, 198) = 0.132, *p* = 0.717, η^2^ = 0.001, sex *F*(1, 198) = 1.852, *p* = 0.175, η^2^ = 0.009, and age *F*(1, 198) = 2.311, *p* = 0.130, η^2^ = 0.012 had no impact.

**Perceived Criminality:** There was not a significant main effect of modality, *F*(2, 198) = 1.108, *p* = 0.332, η^2^ = 0.011, victim status, *F*(1, 198) = 0.355, *p* = 0.552, η^2^ = 0.002, or in the interaction thereof, *F*(2, 198) = 0.678, *p* = 0.509, η^2^ = 0.007. Age had a significant impact *F*(1, 198) = 16.119, *p* < 0.001, η^2^ = 0.073. Past victimisation *F*(1, 198) = 0.195, *p* = 0.659, η^2^ = 0.001 and sex *F*(1, 198) = 0.608, *p* = 0.436, η^2^ = 0.003 had no impact.

**Anticipated Harm:** There was not a significant main effect of modality, *F*(2, 198) = 0.341, *p* = 0.712, η^2^ = 0.003, victim status, *F*(1, 198) = 0.930, *p* = 0.336, η^2^ = 0.005, or the interaction thereof, *F*(2, 198) = 1.377, *p* = 0.255, η^2^ = 0.014. Past victimisation *F*(1, 198) = 0.812, *p* = 0.369, η^2^ = 0.004, sex *F*(1, 198) = 0.057, *p* = 0.811, η^2^ = 0.000, and age *F*(1, 198) = 0.117, *p* = 0.733, η^2^ = 0.001 had no impact.

#### 4.4.2. How Do Demographics and Personality Traits Predict Offence Judgements?

Three linear regression models predicted judgements across offence modality (NCSII, DA, SA), where psychometric (SRP4, BJW, GRBS) and demographic (age, sex) factors were predictors. As victim status did not impact judgements in the previous analyses, analyses were conducted on averaged data, with the past victimisation variable being excluded for similar reasons. Standardised regression estimates are presented in Table 7.

**NCSII:** The model for victim blaming, R^2^adj = 0.270, *F*(8, 59) = 4.097, *p* = 0.001, was statistically significant, wherein higher lifestyle scores (β = −0.083, *p* = 0.015) and feminist beliefs (β = −0.054, *p* < 0.001) predicted less blame. The models for perceived criminality, R^2^adj = 0.037, *F*(8, 59) = 1.321, *p* = 0.251, and anticipated harm, R^2^adj = 0.056, *F*(8, 59) = 1.498, *p* = 0.178, were not statistically significant.

**DA:** Models for victim blaming, R^2^adj = 0.194, *F*(8, 64) = 3.166, *p* = 0.004, and perceived criminality, R^2^adj = 0.259, *F*(8, 64) = 4.152, *p* < 0.001, were statistically significant. Higher age (β = 0.019, *p* = 0.029), affective facet scores (β = 0.089, *p* = 0.008), and lower feminist beliefs (β = −0.029, *p* = 0.029) predicted higher blame; older age (β = −0.053, *p* < 0.001) predicted lower perceived criminality. The model for anticipated harm, R^2^adj = −0.028, *F*(8, 64) = 0.754, *p* = 0.644, was not statistically significant.

SA. Models for anticipated harm, R^2^adj = 0.463, *F*(8, 57) = 8.010, *p* < 0.001, and perceived criminality, R^2^adj = 0.150, *F*(8, 57) = 2.429, *p* = 0.025) were statistically significant. Higher feminist beliefs (β = 0.024, *p* = 0.022) and lower antisocial facet scores (β = −0.141, *p* < 0.001) predicted greater anticipated harm, and lower BJW (β = −0.028, *p* = 0.044) and antisocial facet scores (β = −0.149, *p* = 0.003) predicted greater perceived criminality. The model for victim blaming, R^2^adj = 0.062, *F*(8, 57) = 1.537, *p* = 0.165, was not statistically significant.

## 5. General Discussion

Despite long-standing (NCSII) and recent (USII) legislative developments being implemented to safeguard victims of IBSHA, such acts remain a global problem. Through three studies, we explored differences in societal judgements to vignettes depicting NCSII and USII (and later in reference to DA and SA, which act as more established comparators) as a function of victim sex before testing whether judgements differed as a function of the celebrity status of the perpetrator or victim. Finally, we delineated whether said judgements were predicted by personality traits and beliefs previously implicated in judgements of offending behaviour as well as their own proclivity to engage in such behaviour. Below, we address each of our aims before identifying the limitations and implications of this work.

### 5.1. Judgement Differences Between Offence Types

As predicted, in Studies 1 and 2, vignettes depicting NCSII elicited greater victim blame and anticipated victim harm than those depicting USII, with Study 1 also evidencing a greater perceived criminality of NCSII versus USII. Given that victims of NCSII are more likely (yet importantly, not always) to have sent intimate images to the perpetrator prior to them subsequently being non-consensually shared [22], it is possible that this logic might have underpinned our participants’ reporting, and in turn, the disparity observed in our data. Moreover, this finding supports the rationale underpinning Harper et al.’s [35] Beliefs About Revenge Pornography Questionnaire, which contained subscales pertaining to victims being promiscuous and their inability to avoid vulnerable behaviour. The finding related to perceived criminality might reflect more established societal knowledge of NCSII, wherein legislation has been established in the UK for a decade [4] versus USII wherein legislation is emerging (Online Safety Act, 2023 [5]). However, this finding should be discussed in the context of [a] neither offence-type scoring in the upper quartile and [b] scores for USII being diluted by responses to vignettes featuring male victims. Finally, the finding relating to attribution of harm is likely explained by societal norms surrounding USII [25], wherein there is a high prevalence of people either taking [24] or receiving unsolicited intimate images [12]. In such instances, the frequency of its use, and an associated lack of negative consequences thereafter, would likely suggest to the broader public that such behaviour conveys little harm to those who receive USII. Clearly, education is needed as to the contexts wherein IBSHA can occur, the legislation governing it, and associated harms.

In Study 3, despite there being no differences in perceived criminality or anticipated victim harm across offence types, greater victim blaming was associated with NCSII versus SA (with no difference between either offence and DA scores) after controlling for past victimisation, sex, and age. This finding resonates with pilot work by du Mello Gibbard and Fido [64] and, at least in the context of SA (being mindful of the null finding regarding DA), indicates that victims of NCSII are still subjected to harsher societal views. As with other offence-related differences reported in this paper, however, it is important to contextualise this in that overall, for all offence types, low victim blame judgements were expressed. Akin to this, it was both encouraging to see negatively skewed (high) distributions of perceived criminality and attribution of victim harm, which did not differ as a function of offence type. This opposes research that suggests that the physical nature of sexual abuse might lead to lower attributions of harm for victims of IBSHA [65].

### 5.2. The Role of Celebrity Status

Regardless of offence type (i.e., NCSII, USII, DA, SA) or whether the perpetrator or victim of the offence was positioned as a celebrity, celebrity status neither impacted victim blame, perceived criminality, nor attributed victim harm. These findings contradict Fido et al. [36], who found more lenient judgements of sexualised deepfake media for celebrities versus non-celebrities (both those known to us and strangers) using comparable measures, as well as broader findings of celebrities often being viewed negatively by the public [66]. Though these results might partially be explained by participants not being able to resonate with nameless and hypothetical celebrities in our vignettes, to whom they have not developed a parasocial bond [67], it is possible that this could represent a genuine shift in public opinion. Indeed, in light of greater public awareness of IBSHA [10], NCSII-related legislation now being long established in addition to emerging legislative developments (e.g., [5,6], and well-documented cases of IBSHA by the media, it is possible that the general public now exhibit more pro-victim attitudes and overall greater awareness of the harms of IBSHA.

### 5.3. Predictors of Judgements and Proclivity

Frustratingly, the role of demographics, beliefs, and personality traits in predicting judgements of abuse were inconsistent across vignettes and studies. For example, whereas in Study 1, being male predicted greater blame and lower anticipated harm to male victims of NCSII (but not USII), in Study 2, being male predicted *lower* victim blame in vignettes depicting celebrity perpetrators of USII and greater harm in vignettes depicting a non-celebrity perpetrator of NCSII. No effect of sex was observed in Study 3. Across all studies, age was largely mute as a predictor of judgements, save for those pertaining to DA (Study 3), wherein being older predicted greater victim blame and reduced perceived criminality. As predictors of proclivity, however, being older and male were the sole predictors of proclivity to engage in USII. Though not replicated for NCSII, this does support IBSHA more broadly being considered a gendered crime [14,15] wherein, in response to USII specifically, the term ‘dick pics’ was facilitated through the frequent sending of male genitalia to female victims, to which most college students in one sample reported having received [12].

Similarly, results pertaining to the role of so-called dark personality traits (i.e., psychopathy, narcissism, Machiavellianism) in predicting judgements of IBSHA varied across our studies, and in most cases, did not support extant literature in this field. Using the SD3 scale, no facet of the dark personality predicted victim blame, perceived criminality, or anticipated harm across any vignette presented in Study 1. In Study 2, narcissism predicted low perceived criminality and anticipated harm in vignettes, wherein the perpetrator was a non-celebrity engaging in solely USII, and Machiavellianism predicted higher victim blame in vignettes depicting NCSII, regardless of the perpetrator status. To some extent, these data lend support to Pina et al.’s [37] findings associating enjoyment of NCSII with narcissism and Machiavellianism. However, given that measures and designs are not like-for-like, it would be inappropriate to draw conclusions without further data. Of note, opposing Swanek [38] and Karasavva and Forth [11], where narcissism predicted engagement in NCSII, and Morelli et al. [39], where Machiavellianism predicted engagement in USII, neither were significant predictors here (Study 2). Although, as expected, psychopathy did predicted proclivity to engage in NCSII (see [11,38]).

Regarding judgements, unlike much of Fido’s work, whereby psychopathy and subscales thereof consistently predicted more lenient judgements of NCSII [33,35,48]—as well as deepfake media production [36] but not upskirting [34]—the results of Study 2 in this paper showed only weak and inconsistent associations with victim blame in some but not all vignettes across both NCSII and USII when measured using the SD3 scale. In an attempt to overcome a limitation of the SD3 scale not being able to comprehensively capture subtle variations in the subcomponents of its underpinning constructs, the SRP-4 scale was used in Study 3. Here, though the interpersonal and affective facets did not predict judgement variation in IBSHA (though they did predict greater victim blame for DA), higher scores on the antisocial facet predicted lower anticipated harm for IBSHA (and interesting higher perceived harm for SA), and the lifestyle facet predicted lower perceived criminality for IBSHA. This disparity of results might reflect the SD3 primarily capturing variation in the affective and antisocial (but not interpersonal nor lifestyle) facets of psychopathy. As such, there is seemingly a need for a systematic literature review to fully understand the role of the dark triad in predicting variation in the judgements of and proclivity to engage in IBSHA.

Unexpectedly, both cultural orientation (Study 1) and beliefs about a just world (Study 3) failed to comprehensively predict judgements of IBSHA. While collectivist orientations did yield statistically significant responses, this facet was only associated with greater perceived criminality and anticipated harm in vignettes featuring a female victim of NCSII. To some extent, this maps on to findings depicting those from collectivist cultures as assigning more punitive responses to perpetrators [51]; however, given the lack of associations with victim blame, this result opposes similar findings pertaining to victims of crime [52]. This disparity might be explained by participants being primarily recruited from the UK, which is arguably an individualistic society, and so it would be interesting to explore internationally. BJW did not predict IBSHA judgements, which is surprising given meta-analytic work suggesting that BJW strongly predicts negative victim attitudes more broadly [68]. This finding also contradicts Harper et al. [35], who found BJW to be associated with offence minimisation, viewing victims as being promiscuous and engaging in avoidable behaviour, views where NCSII resulted in minimal harm and was not criminal in nature, and proclivity to engage in NCSII. Of interest, in Study 3, high BJW did predict lower perceived criminality in SA vignettes, which might open up discussions as to how beliefs differentially impact perceptions of acts of image-based and physical sexual harassment and abuse, which warrants further exploration.

Finally, Study 3 also explored the role of beliefs about gender norms. Interestingly, though higher feminist-aligned beliefs predicted less victim blame across DA and SA, they were associated with higher victim blame in cases of NCSII. Though the primary findings lend support to studies that depict sexist-aligned views being associated with greater blame attribution [45,46,47], it is unclear why feminist-aligned beliefs would so strongly be associated with greater victim blame in cases of NCSII, especially when qualitatively opposing views (conservative values) have been associated with anti-victim judgements in NCSII contexts elsewhere [35]. Given that feminist-aligned views were also associated with higher anticipated harm in SA and not DA and NCSII vignettes, this might indicate an interaction in the modality of abuse (as seen in [64]) and further supports the need for further societal education on the impacts of IBSHA and the underpinning scenarios wherein one might become a victim thereof.

## 6. Limitations

The results of this manuscript should be contextualised within limitations. First, though derived solely from the UK to control for legislative variation, with IBSHA being a global problem [7], there is a need to replicate this programme of research internationally. Though IBSHA perpetration has been studied cross-culturally [69], to date, international judgement of IBSHA victims have only been quantified in Fido et al.’s [48] comparison of Norwegian and British participants, wherein British participants perceived NCSII to have more severe impacts on victims. Second, though our vignette and psychometric measures are validated and extensively used in community-based research, it remains possible that participants exhibited social desirability bias. This could also stem to proclivity measures, despite there being variance within the data. Implementing a social desirability response measure (see [70]) could be advantageous. Third, as with all cross-sectional designs, it is impossible to determine causal relationships. Thus, there remains a need for both longitudinal data of predictors of proclivity as well as retrospective qualitative work with those convicted of IBSHA. Fourth, despite adding unique knowledge as to the (lack of) involvement of celebrity status for perpetrators and victims of IBSHA in judgement scores, none of our studies included a condition where both parties held celebrity status, and so they did not truly reflect recent cases such as those involving Stephen Bear and Laurence Fox. In practice, dual celebrity involvement might evoke more tribal opinions, especially given that Internet-mediated abuse on celebrities is considered *part and parcel* of being famous [71].

## 7. Implications of Findings

With an international movement towards implementing more comprehensive IBSHA legislation, these data have practical implications. First, the data indicate that types of IBSHA that have historically and more comprehensively been legislated against convey an understanding of criminality and harm to the public. Though not causal in nature, this might indicate a benefit for the societal awareness of IBSHA to help educate those at risk of perpetration and/or victimisation. This might take the form of educational materials embedded into curricula, media engagement, and targeted interventions. Positively, of course, though variation in victim blame was identified, this was low overall in nature, indicating a societal shift. The data also indicate a seemingly low (if at all) presence of an impact of celebrity status for cases involving perpetrators or victims of such status. Though positive for trial contexts, given that pretrial publicity has been thought to impact jury verdicts, wherein one has greater access to the lives and history of celebrities [72], a next step would be to investigate this complication more thoroughly and to mitigate against any found effects. Finally, for practitioners directly, knowledge of demographic variation in the prediction of USII proclivity can help to inform risk assessment tools and interventions relevant to this crime.

## 8. Conclusions

To conclude, over three studies, this paper tested differences in judgements of NCSII and USII and explored predictors (demographics, personality traits, beliefs) thereof as well as how such variables predicted one’s proclivity to engage in IBSHA. Moreover, the role of celebrity status (across both perpetrators and victims of IBSHA) was explored, which considers public-facing accounts and experiences presented in mainstream media. Finally, our results pertaining to IBSHA were compared with comparable measures of judgements made towards physical domestic abuse and sexual abuse to help contextualise the direction of this literature.

## Figures and Tables

**Table 1 behavsci-14-01021-t001:** Descriptive statistics for outcome judgements by condition.

	Female Victims		Male Victims	
	Non-Consensual Dissemination of Intimate Images	Unsolicited Sending of Sexual Images	Non-Consensual Dissemination of Intimate Images	Unsolicited Sending of Sexual Images
Victim blame	3.528 (0.132)	1.708 (0.142)	3.577 (0.135)	1.875 (0.133)
Perceived criminality	4.706 (0.168)	3.778 (0.180)	3.955 (0.172)	2.547 (0.169)
Anticipated harm	6.324 (0.149)	4.758 (0.159)	5.741 (0.152)	3.616 (0.150)

Note. Figures represent estimated marginal means with ±1 standard error in parentheses.

**Table 2 behavsci-14-01021-t002:** Standardised regression coefficients predicting perceptions of victim blame, perceived criminality, and anticipated harm (by vignette condition).

	Victim Blame				Perceived Criminality				Anticipated Harm			
	F-N	F-U	M-N	M-U	F-N	F-U	M-N	M-U	F-N	F-U	M-N	M-U
Machiavellianism	0.161	0.236	0.361	0.180	0.142	−0.061	−0.076	−0.003	0.305	−0.065	0.153	0.033
Narcissism	−0.075	0.146	0.164	0.060	0.133	−0.030	0.165	−0.114	−0.027	−0.060	−0.126	−0.141
Psychopathy	−0.149	0.067	−0.196	−0.136	−0.158	−0.064	−0.215	0.252	−0.005	−0.102	−0.244	0.124
Horizontal individualism	0.136	−0.239	0.123	−0.002	0.092	0.021	−0.031	0.009	−0.085	0.014	0.018	0.137
Vertical individualism	−0.088	−0.241	−0.183	−0.128	**−0.311**	−0.078	0.136	0.114	0.095	−0.042	0.070	0.163
Horizontal collectivism	−0.119	−0.129	−0.055	−0.029	**0.316**	0.121	0.059	0.014	**0.375**	0.347	0.189	0.258
Vertical collectivism	0.221	0.270	0.151	0.139	0.197	−0.045	−0.150	0.216	−0.089	−0.269	−0.193	0.178
Sex	0.229	0.300	0.022	0.362	0.008	0.034	−0.268	−0.066	−0.255	0.192	−0.158	−0.182
Age	−0.003	−0.028	**0.568**	0.031	0.025	0.319	−0.171	−0.290	−0.054	0.074	**−0.423**	−0.015

Note. The codes ‘F-N’, ‘F-U’, ‘M-N’, and ‘M-U’ refer to the vignette conditions, where F and M indicate whether the victim was either female or male and N and U indicate whether the type of IBSHA was the non-consensual dissemination of intimate images or unsolicited sending of sexual images. The ‘Sex’ predictor was coded as follows: 0 = female, 1 = male. Figures represent standardised beta values. Statistically significant predictors are highlighted in **bold** typeface. Data for this and all studies are available here: https://osf.io/27uty/?view_only=03c41479e43e495fa5ad1b35762ddb9d, accessed on 27 September 2024.

**Table 3 behavsci-14-01021-t003:** Descriptive statistics for victim blaming, criminality judgements, and anticipated harm by condition.

	Non-Celebrity Perpetrators		Celebrity Perpetrators	
	Revenge Pornography	Unsolicited Images	Revenge Pornography	Unsolicited Images
Victim blame	3.52 (0.14)	1.28 (0.15)	3.38 (0.15)	1.59 (0.14)
Perceived criminality	5.91 (0.16)	5.77 (0.16)	6.21 (0.16)	5.78 (0.16)
Anticipated harm	6.13 (0.15)	5.12 (0.15)	6.18 (0.15)	5.38 (0.15)

Note. Figures represent estimated marginal means with ±1 standard error in parentheses.

**Table 4 behavsci-14-01021-t004:** Standardised regression coefficients predicting perceptions of victim blame, criminality, and anticipated harm (by vignette condition).

	Victim Blame				Perceived Criminality				Anticipated Harm			
	C-R	C-U	N-R	N-U	C-R	C-U	N-R	N-U	C-R	C-U	N-R	N-U
Psychopathy	**−0.37 ***	0.20	0.09	**0.32 ***	−0.08	−0.04	−0.14	0.06	−0.11	−0.07	−0.20	−0.19
Narcissism	0.15	−0.03	−0.13	−0.08	0.22	−0.04	0.18	**−0.40 ****	0.31	−0.20	0.12	**−0.39 ***
Machiavellianism	**0.50 ****	0.06	**0.35 ****	0.14	−0.27	−0.19	−0.01	0.07	−0.25	−0.04	−0.05	0.06
Age	0.24	−0.02	0.20	**0.37 ****	−0.16	−0.18	−0.03	0.18	−0.05	−0.27	−0.16	−0.14
Sex	−0.03	**−0.35 ****	−0.10	−0.16	0.06	0.21	0.18	−0.19	0.15	−0.04	**0.30 ***	−0.17

Note. The codes ‘C-R’, ‘C-U’, ‘N-R’, and ‘N-U’ refer to the vignette conditions, where C and N indicate whether the perpetrator was a celebrity or non-celebrity and R and U indicate whether the type of IBSHA was NCSII or USII. The ‘Sex’ predictor was coded: 0 = female, 1 = male. Figures represent standardised beta values. Statistically significant predictors are highlighted in **bold** typeface. * *p* < 0.05, ** *p* < 0.01.

**Table 5 behavsci-14-01021-t005:** Standardised regression coefficients predicting proclivity to engage in NCSII and USII.

	NCSII				USII			
Predictor	β	*T*	*p*	95% CI (β)	β	*t*	*p*	95% CI (β)
Psychopathy	**9.27**	**3.85**	**0.017**	**[1.69, 16.86]**	0.29	0.16	0.873	[−3.29, 3.88]
Narcissism	−1.25	3.69	0.736	[−8.51, 6.02]	1.42	0.81	0.418	[−2.03, 4.87]
Machiavellianism	−1.75	3.46	0.613	[−8.57, 5.07]	2.96	1.80	0.074	[−0.29, 6.20]
Age	−0.03	0.28	0.918	[−0.58, 0.52]	**0.27**	**2.06**	**0.041**	**[0.01, 0.53]**
Sex	4.16	4.29	0.333	[−4.28, 12.61]	**−5.17**	**−2.56**	**0.011**	**[−9.14, −1.19]**

Note. The ‘Sex’ predictor was coded as follows: 0 = male, 1 = female. β refers to the standardised regression coefficient. Significant predictors are presented in **bold** typeface.

**Table 6 behavsci-14-01021-t006:** Descriptive statistics for victim blame, perceived criminality, and anticipated harm by condition.

	Celebrity Victims			Non-Celebrity Victims		
	IBSHA	DA	SA	IBSHA	DA	SA
Victim Blame	2.090 (0.905)	2.140 (0.942)	1.722 (0.704)	2.320 (1.108)	1.641 (0.850)	1.708 (0.905)
Perceived Criminality	6.148 (0.964)	6.108 (0.916)	6.269 (1.054)	5.729 (1.171)	6.154 (0.976)	6.156 (1.053)
Anticipated Harm	6.722 (0.454)	6.529 (0.696)	6.792 (0.366)	6.578 (0.708)	6.654 (0.779)	6.533 (0.694)

Note. Figures represent estimated marginal means with standard deviations in parentheses.

**Table 7 behavsci-14-01021-t007:** Standardised regression coefficients predicting perceptions of victim blame, perceived criminality, and anticipated harm (by vignette condition).

	Victim Blame			Perceived Criminality			Anticipated Harm		
	IBSHA	DA	SA	IBSHA	DA	SA	IBSHA	DA	SA
Interpersonal Psychopathy	−0.007	−0.216	−0.107	0.225	0.067	0.166	0.091	0.228	0.022
Affective Psychopathy	0.018	**0.392 ****	−0.080	−0.194	−0.140	−0.082	−0.302	−0.223	−0.180
Antisocial Psychopathy	0.018	−0.032	0.171	−0.160	−0.011	**−0.452 ****	**−0.272 ***	−0.028	**0.551 *****
Lifestyle Psychopathy	**−0.366 ***	0.185	0.153	0.252	−0.069	0.044	**0.460 ****	−0.102	0.227
Gender Norms	**0.458 *****	**−0.271 ***	**−0.341 ***	0.074	0.131	0.114	0.093	−0.047	**0.250 ****
Belief in a Just World	−0.155	0.000	0.061	0.254	−0.035	**−0.262 ***	−0.009	−0.111	−0.184
Sex	−0.087	−0.098	0.147	0.060	−0.085	−0.145	−0.071	0.042	0.000
Age	0.196	**0.266 ***	0.149	0.093	**−0.618 *****	−0.076	−0.040	−0.191	0.017

Note. The ‘Sex’ predictor was coded as follows: 0 = male, 1 = female. Figures represent standardised beta values. Statistically significant predictors are highlighted in **bold** typeface. * *p* < 0.05; ** *p* < 0.01; *** *p* < 0.001.

## Data Availability

The data presented in this study are openly available in https://osf.io/27uty/?view_only=03c41479e43e495fa5ad1b35762ddb9d, accessed on 27 September 2024.

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
