# Peer review of "Judgement Differences of Types of Image-Based Sexual Harassment and Abuse Conducted by Celebrity Perpetrators and Victims"

_behavsci, 2024, doi:10.3390/bs14111021_

Round 1

Reviewer 1 Report

Comments and Suggestions for Authors

This was a fascinating and very thorough paper, which advances the understanding on image-based sexual abuse. Your findings are robust and the arguments you make in the general discussion are plausible and well-cited. I'm happy to accept the paper in its current form.

Author Response

Thank you for your kind comments and immediate acceptance. 

Reviewer 2 Report

Comments and Suggestions for Authors

I sincerely enjoyed reading the present paper and believe it makes a significant contribution to the field of IBSHA. The paper offers a novel approach to the phenomenon; One of its key strengths lies in how the study not only approaches the topic from a fresh perspective- e.g. exploring elements such as personality and celebrity status-but also effectively contextualises the findings within the current socio-cultural rhetorical terrains. Furthermore, the paper provides a strong review of the literature, referencing key academics in the field, including Pina, Karasavva, Sleath, and Powell.

An additional strength of the paper is its multi-study design. The inclusion of multiple studies enriches the depth of the research and provides a more complex understanding. By approaching the topic through several studies, the authors offer a good explication of IBSHA, ensuring that their conclusions are drawn from broader data sets. This approach not only increases the reliability of the results but also enables the authors to explore the topic from multiple angles, adding considerable weight to their argument. Each study compliments the previous one, highlighting the paper’s methodological rigour.

That said, I have a few minor suggestions for potential improvement. These suggestions are optional, and I understand that the wordcount constitutes a challenge, but may further enhance the clarity and depth of the manuscript.

·         The sentence “Despite this, victims of IBSHA are frequently attributed blame (Flynn et al., 2023) akin to what is seen in physical sexual abuse cases (Mckinaly & Lavis, 2020)” could benefit from a little bit of elaboration.

·         I particularly appreciated how the paper contextualises the findings in relation to contemporary media culture and the broader socio-political environment. The sentence “Though yet to be explored empirically, beliefs about gender roles might also predict IBSHA-related judgement scores; mapping onto viewpoints reinforcing men’s drive for patriarchal dominance via victimising women (McGlynn & Rackley, 2017; Powell et al., 2024)” is well-placed within the argument. However, you might consider expanding on this point by -very briefly-offering more examples of how gender role beliefs influence public perceptions.

·         The sentence “In the context of IBSHA, it is therefore likely that individuals from collectivist cultures will prescribe greater blame across all parties, and deem said actions to be more criminal in nature, than those from individualistic cultures, especially if they prescribe to collectivist values themselves” could be slightly reframed for clarity-I know what you mean, but it might be good to re-frame it in a way that perhaps makes the hypothesis slightly more inclusive. Breaking it into two shorter sentences might improve its readability.

·         I found the discussion around gender particularly insightful, especially the point about male victims e.g. “This was especially the case when said victims were male.” This contet-sensitive approach to gender dynamics is a strong feature of the paper, and it adds valuable depth and contextual sensitivity to the analysis.

·         The sentence “Finally, the finding relating to attribution of harm is likely explained by societal norms surrounding USII (Hayes & Dragiewicz, 2018)” could benefit from further elaboration. Providing specific examples of societal norms and how they relate to the findings would help clarify this point for readers. Additionally, if word count permits, it would be useful to delve deeper into how these norms influence the study's results.

·         In the concluding section, particularly the statement, “it is possible that this could represent a genuine shift in public opinion... it is possible that the general public now exhibit more pro-victim attitudes and overall greater awareness of the harms of IBSHA,” is compelling. Would it be possible to add a sentence or two further explaining the current media discourses and the current cultural landscape?

Author Response

Thank you for the kind comments throughout this review, we have answered all below, and have provided support for any changes:

  • The sentence “Despite this, victims of IBSHA are frequently attributed blame (Flynn et al., 2023) akin to what is seen in physical sexual abuse cases (Mckinaly & Lavis, 2020)” could benefit from a little bit of elaboration.

Thank you for this point. On page 2, lines 54 onwards, we have elaborated on this in this manner:

“Despite this, victims of IBSHA are frequently attributed blame during their victimization, such as for creating and sharing sexual images in the first instance (Flynn et al., 2023) akin to what is seen in physical sexual abuse cases, with victims blamed for being drunk or dressing provocatively (Mckinaly & Lavis, 2020).”

  • I particularly appreciated how the paper contextualises the findings in relation to contemporary media culture and the broader socio-political environment. The sentence “Though yet to be explored empirically, beliefs about gender roles might also predict IBSHA-related judgement scores; mapping onto viewpoints reinforcing men’s drive for patriarchal dominance via victimising women (McGlynn & Rackley, 2017; Powell et al., 2024)” is well-placed within the argument. However, you might consider expanding on this point by -very briefly-offering more examples of how gender role beliefs influence public perceptions.

Thank you, we also added “, as well as viewing them as objects for sexual gratification more broadly” on lines 106-107 to further compound this point.

  • The sentence “In the context of IBSHA, it is therefore likely that individuals from collectivist cultures will prescribe greater blame across all parties, and deem said actions to be more criminal in nature, than those from individualistic cultures, especially if they prescribe to collectivist values themselves” could be slightly reframed for clarity-I know what you mean, but it might be good to re-frame it in a way that perhaps makes the hypothesis slightly more inclusive. Breaking it into two shorter sentences might improve its readability.

Agreed, on page 3, lines 121-125, this now reads:

“In the context of IBSHA, it is therefore likely that individuals from collectivist cultures will prescribe greater blame across all those involved in the behavior. Moreover, individuals from collectivist cultures may deem said actions to be more criminal in nature, than those from individualistic cultures, especially if they also prescribe to collectivist values themselves.”

  • I found the discussion around gender particularly insightful, especially the point about male victims e.g. “This was especially the case when said victims were male.” This contet-sensitive approach to gender dynamics is a strong feature of the paper, and it adds valuable depth and contextual sensitivity to the analysis.

Thank you for the kind words!

  • The sentence “Finally, the finding relating to attribution of harm is likely explained by societal norms surrounding USII (Hayes & Dragiewicz, 2018)” could benefit from further elaboration. Providing specific examples of societal norms and how they relate to the findings would help clarify this point for readers. Additionally, if word count permits, it would be useful to delve deeper into how these norms influence the study's results.

Thank you, on lines 505-507, I have added the following to expand on this point in relation to our results, specifically:

“In such instances, the frequency of its use, and an associated lack of negative consequences thereafter, would likely suggest to the broader public that such behavior conveys little harm to those who receive USII.”

  • In the concluding section, particularly the statement, “it is possible that this could represent a genuine shift in public opinion... it is possible that the general public now exhibit more pro-victim attitudes and overall greater awareness of the harms of IBSHA,” is compelling. Would it be possible to add a sentence or two further explaining the current media discourses and the current cultural landscape?

Many thanks for this. This was actually a consideration in the writing of the manuscript. We tried a few version, however proceeding in such way, really reduced the bite and impact of the statement. So we have opted to thank you for the consideration but haven’t implemented on this occasion. We did, however, consider your cultural landscape point – and though have not written anything on it for this iteration, have chosen not to for a different reason in that this is an area of unknown. We are, however, looking into this in a separate study, and would not like to speak about a culture that we can’t accurately define at this time point. Cheers for this comment though – it’s great to know that our future work is answering questions that people have!

Reviewer 3 Report

Comments and Suggestions for Authors

Thank you for your submission. This is a well written and interesting manuscript. I particularly appreciated that the introduction is thorough, yet concise. I have two minor comments:

1) It was not clear to me in the discussion which finding in particular could be seen as 'logical'. Upon first reading this sentence I took it to mean that victim blaming was logical, and therefore acceptable, in some situations. Please consider revising to ensure clarity.

2) Please proof read the manuscript. In particular, I noticed a number of spacing issues and that 'SA.' in the 'How do demographics and personality traits predict offence judgements' subsection should be bold. Also, I believe there is a typographical error in the paragraph above - I think 'predicter' should be 'predicted'.

Author Response

Thank you for the kind comments throughout this review, we have answered all below, and have provided support for any changes:

  • It was not clear to me in the discussion which finding in particular could be seen as 'logical'. Upon first reading this sentence I took it to mean that victim blaming was logical, and therefore acceptable, in some situations. Please consider revising to ensure clarity.

This is a fair point and it is important to showcase to readers that this is not our intention. This passage has been updated to read:

“Given that victims of NCSII are more likely (yet importantly, not always) to have sent intimate images to the perpetrator prior to them subsequently being non-consensually shared (Gavin & Scott, 2019), it is possible that this logic might have underpinned our participants’ reporting, and in turn, the disparity observed in our data.”

2) Please proof read the manuscript. In particular, I noticed a number of spacing issues and that 'SA.' in the 'How do demographics and personality traits predict offence judgements' subsection should be bold. Also, I believe there is a typographical error in the paragraph above - I think 'predicter' should be 'predicted'.

Cheers for the typo identification. I have proofed the document, an artifact of it being transferred to a template no doubt – I appreciate your time dedicated to this.